

# History of Research on Cloud Types and Naming of Clouds

Peter Winkler

D-82362 Weilheim, Hechenbergstr. 9, Germany

*Correspondence to*: Peter- Winkler (pu_winkler@t-online.de)





**Abstract.** Otto von Guericke was the first who experimentally demonstrated cloud formation by expanding and cooling of compressed air before 1663. Scientists initially grappled with the very question of why clouds float. Early explanations were hindered by limited theoretical understanding, imprecise instruments, and lack of meteorological knowledge.

First attempts for distinguishing various cloud types arose towards the End of the 18th century. A key turning point came in 1803 when Howard proposed a first systematic scheme using Latin terminology for different cloud types. This effort laid the cornerstone for what would evolve into today's internationally recognized cloud classification system. The way toward an accurate understanding was long and iterative. Early laboratory experiments and rudimentary instruments led to recurring misjudgements that persisted for decades. The advent of manned balloon ascents enabled direct measurements of meteorological parameters in the free atmosphere, though initial findings were compromised by instrumental errors. Over time, as more reliable, bias-free devices became available, scientists could obtain accurate data. New measuring techniques had to be developed for determining cloud altitudes and tracking their motion, thereby enriching the understanding of the atmospheric vertical profiles of temperature and moisture influencing the formation of various cloud types.

Cloud research was further intensified by international cooperation. Beginning in the 1890s, scientists started creating internationally acceptable cloud atlases, despite challenges such as early poor photographs with insufficient contrast. Alternatives, like paintings or cloud watercolours, were even considered to overcome these early hurdles. The intensive observation programs—using daily pilot balloons, kites, and later weather aircraft—allowed researchers to uncover cloud formation processes both in stable and unstable atmospheric conditions. Observations also revealed the discovery of clouds above the tropopause and in the mesosphere. After many national attempts to publish cloud atlases the establishment of the World Meteorological Organization in 1951 paved the way for the modern International Cloud Atlas, published in 1956, which standardized cloud observation practices and naming of cloud types worldwide.

## 1 Preliminary remark

According to our current understanding of atmospheric science, two primary processes lead to cloud formation: uplift processes (such as convection and upgliding) and radiation processes (such as long-wave radiation of water vapour and $CO_2$). When the associated cooling is sufficiently strong to lower the temperature below the dew point, condensation of water vapour results in cloud formation. Which cloud type develops depends on the vertical distribution of temperature and water vapour, as well as the stability or instability of the stratification, which can extend over a limited altitude range or the entire troposphere. Additionally, orographic interactions with the atmosphere induce vertical movements (such as mountain waves, foehn winds, and rivers (Erk, 1898), generating waves, turbulence, or vortices, which further modify cloud shapes according to the degree of atmospheric stability.

Most clouds occur in the troposphere, but under specific conditions, clouds can also form in the stratosphere or mesosphere. Ground-based observation of cloud shapes is limited, particularly because cirrus clouds are only perceived above a certain


optical thickness (Koterba, 2020; Spänkuch, 2022). However, thin (subvisible) cirrus clouds can be climatically relevant, although statements on the frequency are often unreliable (Wang et al., 1994).

The immense variety of cloud shapes and their rapid changes have led to numerous interpretations and explanations prior to systematic observation during the Enlightenment. Extensive atmospheric studies using instruments were necessary to achieve a uniform nomenclature of cloud shapes. Initially, clouds were only observed from the ground, but significant knowledge was gained through manned and instrumented balloon ascents and later aircraft observations. The development of cyclone models, including warm and cold fronts and other pressure patterns in large-scale observation networks, influenced the nomenclature of cloud shapes by linking air masses, large-scale vertical movements, and their stable or unstable stratification to specific areas of large-scale pressure structures.

Some cloud shapes can be used to predict weather development, leading to a particular interest in understanding cloud development with time. This interest extends beyond farmers to include outdoor working craftsmen, military personnel, and seamen. Large-scale airflows can be identified by observing cloud movements, and people have increasingly learned to deduce the sequence of weather events based on cloud types.[1]

While clouds are also depicted in art, artistic representations do not contribute to scientific research. Consequently, this aspect is not addressed here, despite the extensive literature on the subject. Similarly, the relationship between types of precipitation and cloud types is not covered.

In order to understand and explain the various cloud shapes from a physical perspective, a centuries-long development process was required.

The high variability of cloud forms posed a significant challenge to progress, until suitable instruments for laboratory simulations became available resulting in theoretical insight.

- Luke Howard and his successors introduced a phenomenological cloud classification system that remains in use to this day. However, early descriptions of cloud forms were prone to misinterpretations. Logical conclusions could not prevent erroneous inferences if the underlying assumptions were incorrect.

- More precise in-situ measurements of cloud physics required the development of relevant physical and aeronautical technologies such as manned balloons, radiosondes, airplanes and meteorological instruments being free of biases that could be operated at the ground or aloft.

- Organizationally, interested scientists from various countries voluntarily joined together in a variety of meteorological and technological commissions (e.g. the International Commission of Cloud Research through the international Meteorological Commission (IMC) from 1894; and the IPCC within IAMAS since 1953 (MacCracken and Volkert, 2019, p. 124).

---

[1] Transition from Cirrostratus to Altostratus and Nimbostratus toward the warm front. Observations show mountains influence by forming prefrontal foehn, which superimposes this standard expectation (Reye, 1864).

The present work reveals key milestones of this evolution demonstrating how progress from a pure phenomenological perspective to a physical understanding of cloud forms was achieved through experiments, theoretical insights, and instrument development.

**2 Fantasy on clouds**

In meteorology the pre-scientific phase can be interpreted as a magical period in which nature was seen and understood in a pictorially sense before the search for a real, physical explanation began. Culturally speaking, this phase is comparable to childhood.

In the time before enlightenment, natural phenomena were mostly interpreted by religious wisdom and by fantasies. In pictures
of meteorological phenomena at the sky, God's punishing hand coming out of clouds hits the sinner. Battleships, fighting knights, raging horses or dragons and even angels were often fantasized into clouds, as children still like to do today. The constant change in cloud shape stimulated ever new explanations of cloud shapes. Such sheets have come down to us from printers who liked to do business with single-sheet printing (Fig. 1). More details are found in Berns (2019).

Frequently occurring cloud types were given figurative and regionally common names such as: fleecy clouds, in England
mackerel sky; in sailors' language, cirrus called wind trees or cat's tails or mare's tails. There were smallpox clouds (mammatus) in the Orkneys (Köppen, 1887; Clayden, 1905, plate 31).

Misconceptions persisted for a long time: Clouds were believed that they could be given another direction by cannon shots. Clouds were regarded to be rigid bodies. When they rub against each other, electricity is generated (Lampadius, 1806, p. 83).

It first had to be recognized and understood that cloud shapes are formed by a meteorological process and they are constantly
re-forming by air flowing through the cloud (except in ground fog).

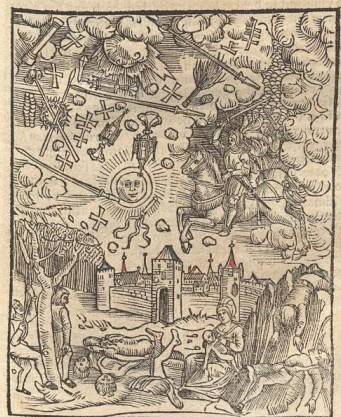

Fig. 1: Frontispiece of J. Grünpeck: *Ein newe außlegung Der seltzamen wundertzaichen* […] (1515), showing a knight on horse and other signs in the clouds as a consequence of exuberant fantasies.

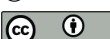

In addition, the scholastic approach, which was still widespread for a long time, only considered the pros and cons of a hypothesis, without a real physical process to be seen as a cause for the phenomenon. A typical view was that in the morning terrestrial vapours would rise with warming of the ground by the sun. When radiated by the sun such terrestrial vapours could increase the atmospheric electricity aloft, enhanced by cooling. Warm vapours would diminish positive atmospheric electricity. Due to the lack of knowledge about the magnitude of the latent heat amount, the warming of the air during heavy dews was

strongly overestimated (Hube, 1790). Such argumentations could only be refuted gradually through systematic experiments.

**3. Early explanations of cloud formation and the floating of clouds – nearly 200 years of the vesicle hypothesis**

In 1667, Robert Hooke (1635-1702) in England proposed a weather observation scheme. He recommended noting the direction of cloud movement and the shape of the cloud base, being either flat or wavy and irregular (Hooke, 1667). He also suggested determining the height of the clouds. However, the scientific community did not pay attention to his suggestions; Hooke was

far ahead of his time.

The next stage focused on the formation of clouds. Otto von Guericke (1602-1686), the inventor of the vacuum pump and mayor of Magdeburg in Germany, conducted numerous experiments and published the results in his seven-part work "Experimenta nova" in 1672 (Guericke, 1672). The manuscript was already completed in 1663 but was not printed until nine years later. In chapter A of book III, he described an experiment to produce a cloud for the first time (Fig. 2): a large vessel

was evacuated and connected to a small vessel containing moist air at atmospheric or higher pressure. When the connecting tap was opened, the air expanding into the vacuum cooled adiabatically, and mist droplets became visible that slowly sank down. He assumed these droplets to be hollow vesicles without further reasoning.

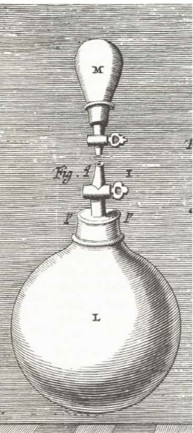

**Fig. 2: Guericke's device for his fog experiment performed before 1663.**




Edmond Halley (1656–1742) in London also developed the vesicle hypothesis in 1690, although it is unknown whether he was aware of Guericke's experiment. Halley imagined that when water boils, vapour bubbles leave the surface as hollow vesicles filled with a light gas due to the heat effect. This makes them specifically lighter and able to float. (Halley, 1690): "*I have formerly attempted to explain the manner of the rising of vapors by warmth, by showing that if an atom of water were expanded*

*into a shell or bubble, so as to be ten times as large in diameter as when it was water, such an atom would become specifically lighter than air [...]."* Atom for him meant particle, not in today's sense as the smallest particle of a chemical element.

Guericke's findings, published in 1672, were partly made known earlier to experts by Caspar Schott (1734-1794), a Jesuit at the University of Würzburg, who corresponded with Guericke and quoted from his letters (Schott, 1664, p. 21 ff.; Heinecke et al., 2019). In 1672, a review of Guericke's work appeared in the Philosophical Transactions of London, which may have

inspired Halley to adopt the vesicle hypothesis. However, the existence of hollow vapour or fog bubbles could not be verified, nor could the nature of the rarefied gas inside.

The young Gottfried Wilhelm Leibniz (1646-1716) studied Guericke's "Experimenta nova" intensely and corresponded with him in 1671 and 1672 (details in Dannemann, 1889). In 1710, he wrote his treatise "De elevatione vaporum," in which he also assumed hollow vesicles that had to be specifically lighter than air to be able to float. Leibniz referred to Otto von Guericke

(Guericke 1672, Chapter IX). Leibniz famously ignored English-language literature and did not mention Halley's arguments. His theory was considered correct for another 170 years.

Christian Wolff (1679-1754), the philosopher who influenced the development of enlightenment all over Europe with his ideas, observed droplets appearing white when they crossed a beam of light in a dark room. By analogy with white foam on brown beer, which consisted of bubbles, he concluded that the floating droplets must also be hollow vesicles (Wolff, 1722, § 84).

In 1743, Christian Gottlieb Kratzenstein (1723-1795) submitted a treatise on the physical prize question to the Académie des Sciences de Bordeaux on "L'élévation des vapeurs et des exhalations" (Kratzenstein, 1743; Kratzenstein, 1744). He confirmed the vesicle hypothesis with the argument that a rainbow can never be observed in a cloud, and therefore cloud droplets cannot be compact but must be hollow. His paper received widespread attention. Kratzenstein saw another reason for the rise of vapour vesicles in the adhesion of the surrounding air molecules. At low altitudes with high air pressure the cohesion of the

surrounding air molecules should be strong. The vesicles would therefore receive upward buoyancy. At higher altitudes, the cohesion of the air molecules decreases due to falling air pressure, causing the buoyancy to come to a standstill, and the vapour vesicles float at this altitude (Fig. 3). Kratzenstein overestimated the size of air molecules in relation to water vesicles ($4*10^{-10}$m instead of $10*10^{-6}$m as assumed by Kratzenstein). Additionally, Kratzenstein only knew refraction of light, not diffraction on very small particles, which was discovered much later by Fraunhofer (1825). Furthermore, Kratzenstein did not understand

the vapour pressure variation and its dependence on the radius of curvature of a small droplet. Nonetheless, the specious logic of his argument convinced many scholars.

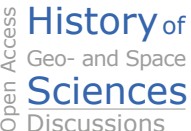

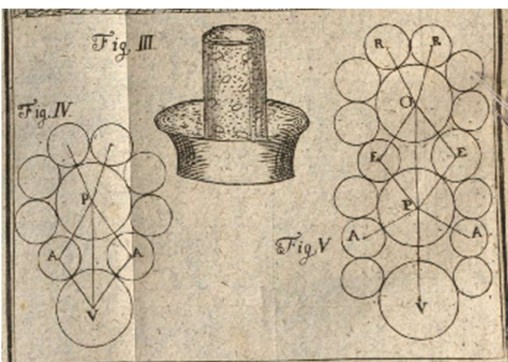

**Fig. 3: Kratzenstein's consideration of an uplifting motion of water vapour molecules (P) due to the cohesion of air molecules (A) and its reduction with falling atmospheric pressure.**


The dogma of the written word about cloud vesicles had an impact for almost 200 years: no one seriously doubted the cloud vesicles because the explanation seemed perfectly logical. Leading scientists like Euler, Saussure, Humboldt, Volta, Kämtz, and Clausius and others held on it; even the philosopher Arthur Schopenhauer (The World as Will and Representation (Appendix)) had no doubts about the existence of cloud vesicles.

However, over time, doubts were raised about the vesicle hypothesis. One of the first was the physicist Ludwig Wilhelm Gilbert (1769-1824), who translated Howard's essay "Modification of clouds" into German in 1805 and stated in the preface that only water vapour could be found inside bubbles (Gilbert, 1805). Vesicles would therefore have to sink downwards and could not stay suspended. In later textbooks, authors stated again that only water vapour could be inside vesicles, but this did not give rise to particular doubts about their existence. The formation of rain was thought to result from the merging of cloud

vesicles (Schleiss von Löwenfeld, 1861, p. 66 ff.).

In 1847, Augustus Volney Waller tried to detect vesicles under the microscope but could not confirm them, instead he considered them as "spherules" and not as "vesicles" (Waller, 1847). His work was ignored. He had not examined cloud droplets in the atmosphere but rather those created by breathing.

Kuhn (1866) observed a strongly coloured rainbow in a fog and concluded the presence of compact droplets. However, he

argued that compact droplets could coexist with vesicles in order to keep open a backdoor.

Only Kober was able to show through microscopic examinations that vesicles did not exist in the atmosphere (Kober, 1871). His dissertation appears to have had little spread and received barely any attention after publication. The vesicle hypothesis ultimately remained in teaching until 1885.

In that year, Richard Assmann (1845-1918) examined real cloud drops on top of the mountain Brocken in Harz Mountains

with a microscope and found no evidence of the existence of hollow vesicles. This hypothesis was finally refuted with the





publication in the "Meteorologische Zeitschrift," one of the leading journals at that time (Assmann, 1885). Nevertheless, many articles appeared afterward in which the vesicle hypothesis was still uncritically adhered to.

## 4. Early Cloud Classification Approaches

Clouds are constantly changing their forms that make air movements visible. Their shapes depend on the stability or instability
of the air stratification. Radiation and atmospheric dynamics (such as vertical movement and wind shear) also play important roles in forming various shapes. The temperature determines the physical state (liquid or solid phase) of the cloud particles. It is not surprising that many initial cloud names were not physically justified.

The mathematician Friedrich Meister (1724-1788) in Göttingen developed an initial approach to classifying cloud types and he distinguished:

• bulbous, convex shapes
       • holey, concave shapes

He described clouds as round or long, with parts that appeared spongy or grainy. These basic forms could combine into various larger cloud structures, governed by certain atmospheric mechanisms rather than randomness (Meister, 1780).

The Societas Meteorologica Palatina focused only on observing cloud coverage, density, and colour.[2]

In a subsequent approach, Jean Baptiste de Lamarck (1744-1829) distinguished five types of clouds across three atmospheric levels (Lamarck, 1802). He later expanded his system to twelve types. Although the French names for clouds did not gain much attention among scholars, the division of the atmosphere into three levels, which Lamarck introduced, is still used today. The 1939 edition of the International Cloud Atlas speculates on the reasons for this: Lamarck's somewhat peculiar French names may not have been readily adopted in other countries, or the paper might have been discredited by appearing alongside
forecasts based on astrological data. Four of Lamarck's five principal cloud types appear in Howard's naming scheme, but there is no evidence that Howard was aware of Lamarck's research or he was in contact with him.

Luke Howard (1772-1864), a businessman who manufactured pharmaceutical chemicals, observed clouds as a hobby and categorized them into three basic types (Fig. 4). He also suggested cloud symbols for quick notation, which later became indispensable in a modified form when constructing weather maps.

Howard's significant achievement was the introduction of Latin terms, as Latin was still widely used as a scholarly language, giving Howard's system an international appeal. Thus, he succeeded in eliminating the use of local and trivial names. He introduced three basic types and supplemented them with mixed types, distinguishing seven cloud types in total. Howard assumed only two essential height levels, with the middle and upper areas coinciding (Howard, 1803).

---

[2] Ephemerides Societatis Meteorologicae Palatinae 1781, Vol, 1 (1783), p.11.

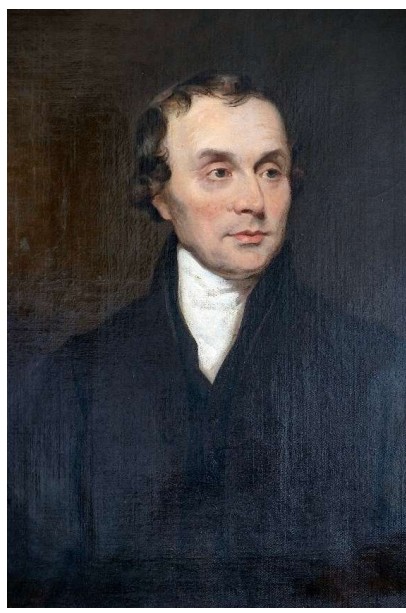

\ Cirrus.
⌒ Cumulus.
‿ Stratus.
\⌒ Cirro-cumulus.
∟ Cirro-stratus.
⌒‿ Cumulo-stratus.
\⌒‿ Cirro-cumulo-status, or Nimbus.


**Fig. 4: Luke Howard, painting by John Opie (credit of the Royal Meteorological Society) and cloud symbols recommended by him.**

It should be noted that although Howard's cloud names were later adopted, they were interpreted in a more differentiated manner. When labelling the shapes, he overestimated the role of electricity in the cirrus level. Howard did not yet understand

the stable and unstable states of stratification and used available knowledge to draw analogous conclusions. The similarity of cirrus shapes to Lichtenberg's dust figures (Fig. 5), generated by static electric fields, and led him to conclude that strong electric fields must also be present at the altitude of the cirrus clouds. The ice crystals would arrange themselves according to the electric fields (Hamanaka, 2015), showing how the "electric fluid" is conducted through the cloud. Even before Howard, Jean André Deluc (1727-1817) noted the similarity between some cirrus species and Lichtenberg's figures (Deluc, 1786,

Section XII). However, Howard was aware that his conclusion was deduced by analogy rather than as a result of an inductive chain of thoughts.

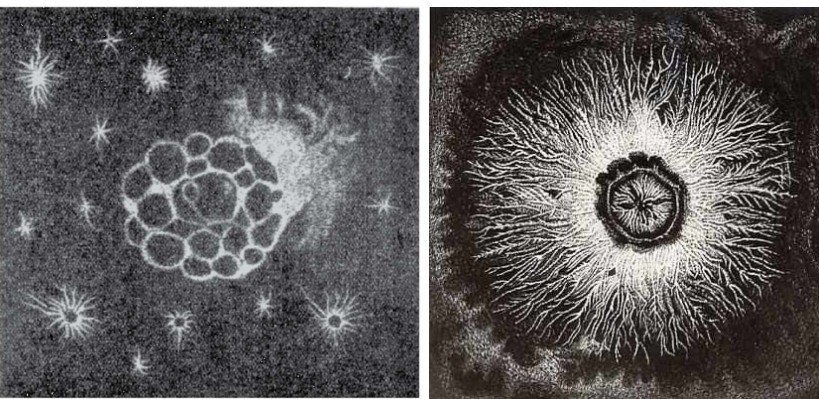

**Fig. 5: Lichtenberg figures. From their similarity to cirrus forms, Howard concluded on a large effect of electricity in the high atmosphere on the cloud shape of cirrus.**


August Wilhelm Neuber (1781-1849) followed the botanical naming system and defined 228 different types of clouds (Fig. 6) (Neuber, 1829). This was unacceptable for meteorologists.

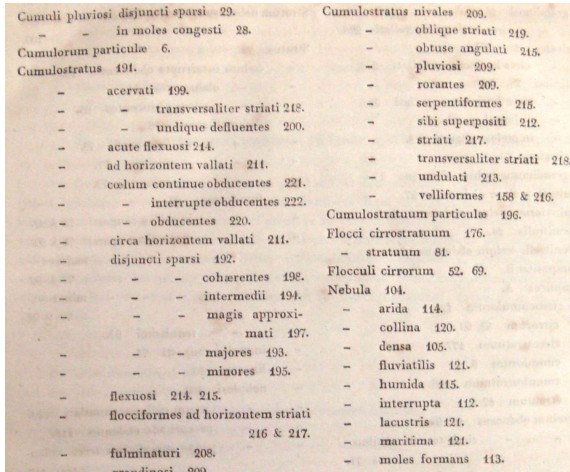

**Fig. 6: Neuber (1829): Example of the naming of stratocumulus variants. Unlike Howard's with only seven cloud types, Neuber**
**proposed 228 cloud types.**

In his Berlin lectures in 1827, Alexander von Humboldt (1769-1859) described a striking parallel orientation of the polar lights and cloud bands at the cirrus level. He noted: "*One often notices really bright clouds. Tienemann*[3] *in Iceland first pointed out that the so-called sheep (Cirrus stratus), which I observed on the Chimboraço at a height of 3-4000 toises above me, may be*

*connected with the Polar Lights. He noticed that they are luminous on some nights*." British naval officer Parry (1790-1855)[4] stated that the morning after an aurora borealis, he saw clouds forming in the direction of the arc, leading to the term "polar bands."[5] The assumption that parallel cirrus bands would align with the magnetic field and point towards the magnetic pole was later abandoned (Dorno, 1925). However, Dorno argued that, in particular, ice plates might systematically align with the sun's rays, and that those warmed by radiation would evaporate while others could cool below the temperature of the

surrounding air and persist. Consequently, radiative effects might partially contribute to the formation of polar bands. This explanation is also unconvincing, as there is no reason for a systematic alignment of ice plates. Eventually, all parallel cirrus bands were referred to as polar bands, as they appear to converge at a "pole" in perspective (Bezold, 1894).

For a long time, cloud shapes were not systematically researched. Observations focused on the degree of cloud cover and determining its climatological averages, without considering cloud types. Dove (1803-1879) noted (1837, p. 48) that a cloud

is constantly forming and disappearing, suggesting clouds should be considered a process rather than a solid object.[6]

**4.1 The role of adiabatic processes**

James Pollard Espy (1785-1860) gave an important impulse to cloud research. He recognized that adiabatic processes occur in the atmosphere due to the conservation of energy. The release of latent heat during condensation drives the formation of thunderstorms and low-pressure areas (Espy, 1841). Espy conducted experiments with a "nephelescope," similar to Guericke's

apparatus, without knowing about it (Fig. 7). He relied on Dalton's results on total pressure as the sum of partial pressures.

---

[3] Dr. med. Friedrich August Ludwig Thienemann (1793-1858), Birdwatcher who spent a year and a half in Iceland in 1820; Allgemeine Deutsche Biographie 38, 1, 1893.

[4] Discoverer who travelled by ship to the northern polar regions in 1818.

[5] Parthey, Gustav: Alexander von Humboldt Lectures on physical geography. November 1827 to April, 1828. Notes by G. Parthey. [Berlin], [1827/28]. [= Transcript of Alexander von Humboldt's 'Cosmos Lectures' at the University of Berlin, November 3, 1827 - April 26, 1828.] German Text Archive (Berlin-Brandenburg Academy of Sciences).

[6] Friedrich Kries had already presented this idea in a similar way in his textbook (Kries, 1821, p. 464), but Dove formulated the matter most succinctly; compare also Fischer, 1808, p. 238.


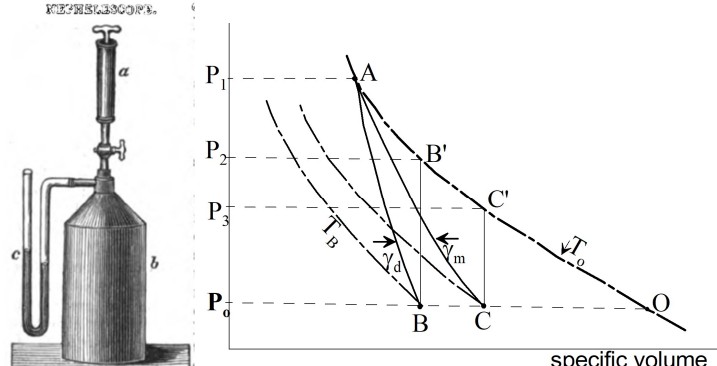

**"Fig. 7, above: Nephelescope according to Espy (1841). Below: Graphic according to McDonald (1963, p. 638), illustrating the temperature and pressure changes in the nephelescope during dry ($\gamma_d$) and moist adiabatic ($\gamma_m$) expansion.**

Espy's vessel b was brought to a higher pressure P1 with the pump a, and temperature equalization with the environment was awaited. After removing the pump and opening the tap, the air cooled dry adiabatically and reached the specific volume B. Once pressure equilibrium prevailed, the tap was closed. As the temperature equalized again (B′), the pressure was increased to P2. With water at the bottom of the vessel, the air could be made saturated after increasing the pressure to P1 again. Upon opening the tap, the air now cooled moist-adiabatically, reaching a higher specific volume due to the release of latent heat up

to specific volume C. The tap was closed again, and temperature equilibration in C′ was awaited. The pressure now increased only to P3. Espy demonstrated in this simple way that cumulus clouds experience additional buoyancy through latent heat release, but his findings did not intensify the study of cloud types at that time.

Fraunhofer investigated scattering and diffraction (Fraunhofer, 1825)[7] on drops and crystals, leading to the conclusions about ice or water clouds. Optical phenomena like coronas, halos, perihelion, or sun pillars provide evidence of cloud drops or ice

crystals. Precise height measurements using photogrammetry revealed the existence of supercooled water droplets or ice crystals in translucent clouds above freezing level, creating coronas or halos around the sun or the moon.

**4. 2 Further Development of Howard's cloud system**

Following this period, individual observers attempted to further develop Howard's cloud system, with Poey and Ley receiving wider attention. This was a learning phase where many misjudgements still occurred.

Poey (1799-1891), a meteorological observer in Havana, proposed two new cloud names to the Societé Météorologique de France in 1863 and defined the fractocumulus type (a wind cloud because it moves quickly) for the first time. He also used additions such as pallio- or pallium (=mantle) clouds (Poey, 1863; Poey, 1874). These names, based purely on appearance

---

[7] See also: Kämtz (1836): Lehrbuch der Meteorologie III, 8. Abschnitt; p. 94 ff. 115f.

without any meteorological basis, were not accepted. Murphy (1870) commented on Poey's suggestion, stating that new cloud names were justified only with a physical explanation of the conditions under which certain types formed (Murphy, 1870).

In England, Clement Ley (1840-1896), rector of St. Peters Church in Ashby Parva, Leicestershire, made cloud observation his life's work (Ley, 1879; Ley, 1882, 1884). His approach was principally physical, though he did not yet understand the variety of atmospheric temperature and humidity stratifications and their effects on cloud shapes. Ley suggested too many names, but only some were retained. For example, he introduced terms such as radiation clouds, interfret clouds, and inclination clouds, based mainly on his interpretation rather than on a physical process in the atmosphere. He also struggled and capitulated with

the huge variability of cirrus clouds. His system for classifying clouds was considered inconsistent (Grossmann, 1894). His book "Cloudland" (Ley, 1894) was published posthumously. However, he had already given the correct term to the species Altocumulus castellatus (todays term castellanus), which indicates instability at altitude and is considered a thunderstorm predictor for the afternoon (Fig. 8). Although its prognostic significance was already known, Ley published the first drawing.

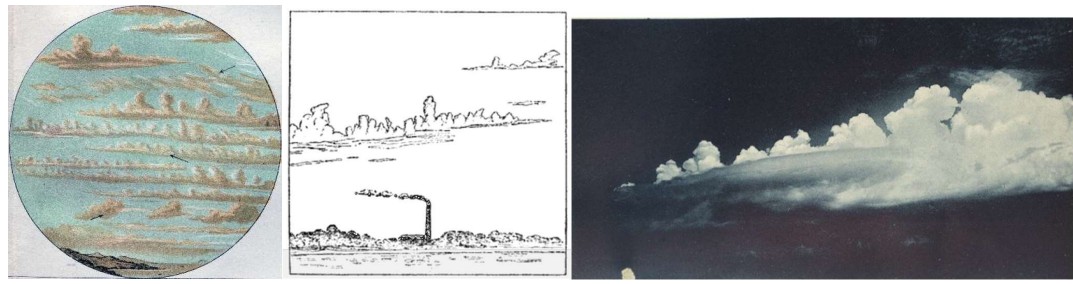


**Fig. 8: Left: Drawing of Altocumulus castellatus by Ley (1879). The tower-shaped convection indicates unstable stratification at altitude, which increases over the course of the day and often leads to thunderstorm formation. Right: Drawing by Köppen (1891) in Hamburg of Altocumulus castellatus in the morning, which was followed by a night thunderstorm. A relatively good photo was achieved by Quervain (1908).**


**Table 1: Nomenclature of Ley (1894) and international nomenclature of cloud forms.**

| Clement Ley | International |
|---|---|
| Nebula | Stratus |
| Stratus Quietus | Strato-Cumulus |
| Stratus Maculosus | Alto-Cumulus |
| Cirro-Macula | Cirro-Cumulus |





| Nimbus | Nimbus |
|---|---|
| – | Alti-Stratus |
| Cirro-velum | Cirro-Stratus |
| Cumulus | Cumulus |
| Cumulo-Nimbus | Cumulo-Nimbus |
| Cirrus | Cirrus |
| Stratus castellatus | Altocumulus castellanus |
| Cirrus mammatus | Cirrus mammatus |

Additionally, Ley named smoke layers or Sahara dust as nebula pulvera, which is not a cloud strictly speaking (Ley, 1894, p. 114f.) and he proposed the term nubes fulgens, although it was unclear if he had actually seen glowing noctilucent clouds. He estimated their altitude to be 91,800 meters.

### 4. 3 Further insights through meteorological research

In 1872, the International Meteorological Committee was founded in Leipzig in Prussia, (Börngen and Foken, 2022), primarily
to standardize instruments and unify observation methods. Interest in clouds grew anew after the meteorological congress in Leipzig in 1872 and subsequent congresses. It was recognized that a morphological cloud classification was necessary to draw conclusions about frequently occurring typical atmospheric states from cloud shapes. Initially, climatological aspects were of primary importance, however. It wasn't until 1876 that observers in Austria were instructed to note main cloud shapes such as cirrus, stratus, cumulus, and nimbus.[8]

One reason for the long-standing lack of interest in cloud shapes could be that an understanding of the meteorological conditions that produce different cloud shapes had to be developed first. Only when this combination was understood cloud shapes could allow conclusions to be drawn about meteorological conditions, such as unstable or stable air stratification over a low or high-altitude range. These conditions could be clarified step by step after the start of scientific balloon flights from 1880 onwards, in which reliable vertical profiles of temperature and relative humidity were measured while flying through

---

[8] Jelinek (1876, p. 127 f.): If recognizable, the direction of motion should be noted, for which a Braun nephoscope (cloud mirror) could be used. There was no method for determining height, except when mountains of known height were visible nearby.

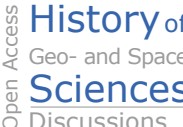
clouds. In 1882, the International Meteorological Commission first demanded information on cloud height, cloud movement, and cloud density. At that time, such tasks could only be performed reliably by specially trained observers.[9]

Cloud movements allowed indirect observation of high-altitude wind speed and direction, for which cloud height had to be precisely determined using photogrammetry techniques. Experiments had already begun in many countries (e.g., at the Kew Observatory in London by Strachey and Whipple, 1891), but uncertainties in the alignment of the optical axes of the two

cameras still had to be overcome.

Over time, it was recognized that unstable air stratification leads to convection, even at limited altitudes, causing cumulus cloud types to form. If there is an inversion above an unstable layer of air, the cumulus cloud does typically not develop further in height but spreads out as a cloud layer at the inversion. In the upper troposphere, when there is sufficient moisture, cirrus clouds form with a wide variety of shapes, also due to the high wind speeds, wind shear, and turbulence occurring there.

The physicist W. von Bezold (1837-1917) warned (Bezold, 1892, p. 18) that one should not proceed too quickly "to create an artificial system for the classification of clouds on purely descriptive grounds, following the method of Linnée's plant scheme, which does not consider the formation and dissolution of clouds." He believed that this approach would not be suitable for promoting the understanding of any atmospheric cloud phenomenon. At that time, he considered meteorology to be an empirical science, but with his warning, he intended to emphasize that cloud types must be interpreted in connection with the

physical conditions of their formation. Nevertheless, significant progress had already been achieved.

An important observation during early balloon flights was the coexistence of supercooled water drops and snow crystals at altitudes with temperatures below freezing. Pilots were able to confirm the presence of hydrometeors in both states through direct observation. Alfred Wegener (1880-1930) emphasized that ice particles have a lower vapour pressure and cannot coexist with cloud droplets for long because snow crystals will grow at the expense of supercooled water droplets (Wegener, 1911).

The development of the Bergen cyclone model by Jakob Bjerknes (1897-1975) provided another important impetus for understanding cloud types (Bjerknes, 1911). The idealized cyclone model described the circulation and mixing of cold and warm air masses in a vortex and the associated large-scale vertical movements, with up-sliding at the warm front and the advance of cold air on the cold front, which pushes itself under the warm air and leads to the formation of cumulonimbus clouds (Fig. 9). By this concept "air mass thinking" was introduced into meteorology in general and into cloud research in

particular. This meant taking into account, in particular, the temporal variation in the displacement of air masses and their impact on cloud formation. New ideas could be developed where in the cyclone area different cloud types preferentially can be expected: stratiform clouds at the cirrus level, thickening to altostratus with a lowering cloud base and finally turning into nimbostratus with rain. This sequence is expected before the warm front. When denser air behind the cold front pushes underneath the warm air, high-reaching cumulus clouds develop, leading to the formation of showers. In the warm sector

stratiform clouds should prevail.

---

[9] Bericht des Int. Met. Komitees 1885 (Hamburg, 1887, Annex III).

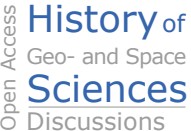
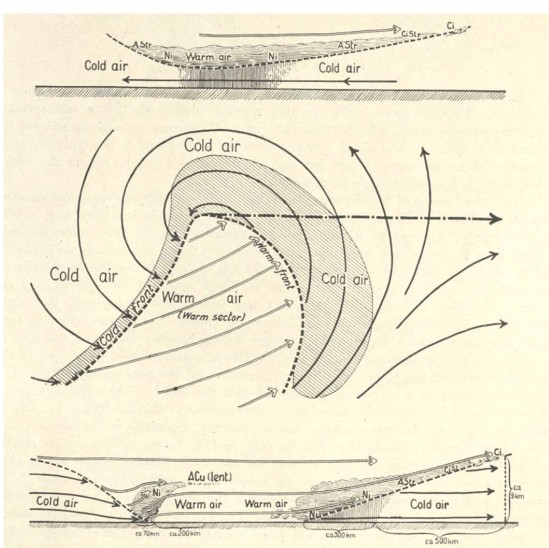

**Fig. 9: Bergen cyclone model (Bjerknes, 1919) with typical cloud cover on warm and cold fronts and north of the pressure minimum centre: at the warm front, layered clouds form as the warm air gradually slides upwards; at the cold front, high-reaching cumulus**
**clouds with showers and gusty winds form as the advancing cold, dense air suddenly pushes the warm air upwards.**

In high-pressure systems, sinking air warms adiabatically, creating a sinking inversion. This also induces large-scale effects on cloud cover. In the eastern and cold part, shallow fair-weather cumulus clouds often occur due to intense radiation and ground-level instability extending to the subsidence inversion. The western warm section of the high has few clouds. The
transition area is typically marked at altitude by a long cirrus band.

At the meteorological congress in Munich in 1891, the first cloud classification was agreed upon based on the suggestions of Abercromby and Hildebrandsson (Abercromby, 1887a).

Many modifications of the basic cloud types arise from interactions with orography. When a mountain ridge is overflowed, leeward waves create lens-shaped clouds.
A special feature is the mammatus type, from which downward instability can be diagnosed because cool, water vapour-saturated air lies above warm and dry air, and convection develops downwards. This cloud shape often forms at the lower edge of the cirrus anvil of a thundercloud. As long as layering stability was ignored, observers found it difficult to correctly determine the mammatus type of clouds. Schultz et al. (2006) show in a review that even today not all questions have been answered. The reason for the smooth surface of mammatus clouds is still unclear. The classic explanation was described by
Osthoff (1906). Doswell (2008) points to an analogy in oceanography, where the difference in the diffusion rate of heat and



salinity leads to the formation of so-called salt fingers. In the atmosphere, there could be a similar difference in the rate of diffusion of heat and ice crystals. Further research seems to be necessary here.

Cirrus clouds are particularly diverse due to the Kelvin-Helmholtz instabilities that often occur at this altitude and the resulting turbulence from overturning waves (e.g., Quante, 2006).

When a cumulus cloud rapidly pushes up an overlying moist layer, a cap will develop (Fig. 24). Some properties relate exclusively to a specific cloud type. Common and scientifically unsubstantiated names were discarded by the international community.

**5. Prognostic use**

The evolution of cloud observation for weather forecasting was fascinating forever. Researchers like Forster (1789-1860) and

Ley made the first steps for understanding how cloud formations can indicate impending weather changes. For example, the appearance and thickening of cirrus clouds can signal an approaching precipitation area, as noted by Forster in the early 19th century (Forster, 1813).

In England, an attempt was made in 1880 to use cirrus cloud observations for forecasting, following a suggestion by Ley. Abercromby (1842-1897) published another attempt to use cloud types for weather forecasting (Abercromby, 1885). He

compressed his observations into two diagrams (Fig. 10), the first showing a low-pressure area, albeit without warm and cold fronts; the second diagram included the cloud types and the typical direction of movement of a low-pressure system over England.

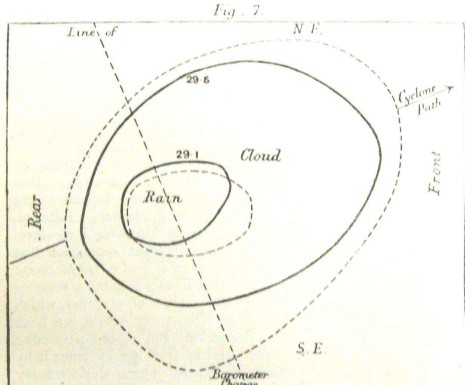

**Fig. 10: Typical cloud types and other weather phenomena relative to the centre of a low-pressure area and its direction of movement**
**(Abercromby, 1885).**

Altocumulus castellanus clouds, identified by Ley, are reliable indicators of atmospheric instability, which often leads to thunderstorms some hours later (above, Fig. 8) (Quervain, 1904). Abercromby, on the other hand, focused on classifying cloud types to enhance weather forecasting accuracy in the late 19th century. His work highlighted the typical cloud formations and

their movements in low-pressure areas over England.

The need for more research on atmospheric conditions that influence cloud formations was emphasized by the Meteorological Committee in Zurich in 1888. Over time, the transition of cloud types, such as from Cirrostratus to Altocumulus and Nimbostratus, helped meteorologists expect the approach of warm fronts in low-pressure systems.

The International Cloud Atlas of 1930 and 1939 presented typical cloud conditions associated with low-pressure systems (Fig.

11) to inform observers adequately.[10]

Overall, these early contributions have significantly shaped modern meteorology and weather forecasting.

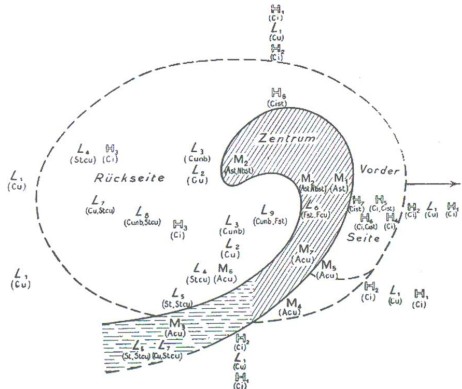

**Fig: 11: In the International Cloud Atlas (1930), another representation of the typical cloud conditions in a low-pressure system was shown and reprinted in the following edition of 1939, p. 47).**

## 6. Influence of orography

As air ascends while flowing over mountainous terrain, cloud formation in the form of a cloud cap can occur if the contained water vapour reaches the saturation point. In the lee of high mountains, stationary banner clouds or so-called parasitic clouds can develop (Lampadius, 1806, p. 134; Kober, 1871, p. 398), although the conditions for their formation were misunderstood for a long time. Mountain ridges can generate waves in the lee, at the maxima of which lenticular clouds form, provided the

water vapour exceeds the saturation point.

---

[10] Bericht des Int. Met. Komitees 1885 (Hamburg, 1887), Annex III: Report by Scott, Secret. of the Commission and head of the Meteorological Service in London.

Mountains promote the formation of thunderstorm clouds (Börnstein, 1901). In valleys, if rotor-like cross-circulation is induced, one slope may be cloud-covered while the other remains cloud-free (Börnstein, 1901). Slope clouds have not been specially named. In 1889, Mohorovičić described the observation of a stationary rotor cloud at the coast of Croatia (Fig. 12) (Mohorovičić, 1889).

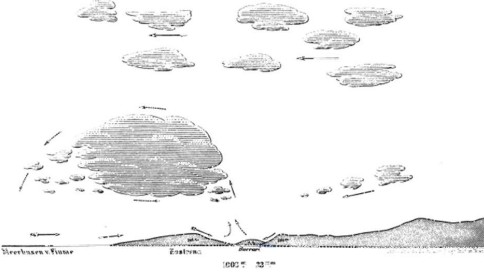


**Fig. 12: Rotor cloud in the lee of the Yugoslav Adriatic coast.**

Observations from both, balloons and airplanes have confirmed that a river course can become visible in a closed cloud cover (Erk, 1898) or even cause its dissolution (Fig 13), (Anonymous, 1917, Fig. 127).

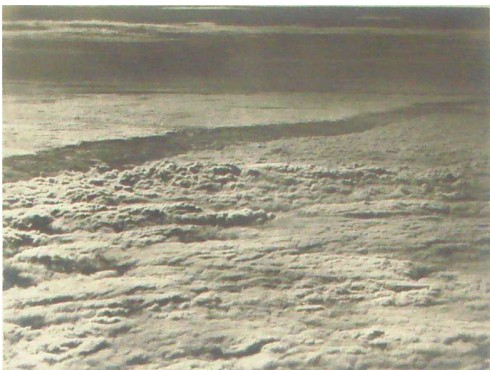


**Fig. 13: Aerial photograph of a closed cloud cover (Anonymous, 1917) that has dissolved over a river course.**

## 7 International Commission for Cloud Research

The International Meteorological Committee decided during the meeting in Uppsala in 1894 to establish the International Commission for Cloud Research to prepare for the International Cloud Year 1896/97. The members included Hildebrandsson (Sweden), Riggenbach (Switzerland), Neumayer (Hamburg/Germany), and Abercromby (England). In 1895, Sprung (1895)




reported on the recommendations, which emphasized the importance of determining cloud height and direction of the cloud movement through corresponding photographs.

Scientists from 11 countries participated in this observation program, for which specially trained scientists were required. In later sessions, there was a desire to disseminate information on cloud movement in weather telegrams, but this intention had to be postponed again in 1907.

This historical consideration reflects the collaborative efforts of meteorologists from different nations to enhance the understanding of clouds and their movement.[11]

## 7. Further Research

As mentioned above, Espy described in 1841 the importance of the release of latent heat during condensation for enhancing convection. Robert Mayer's (1814-1878) work laid the ground for a deeper scientific investigation of weather phenomena. The first law of thermodynamics, also known as the law of energy conservation, states that energy cannot be lost, only transformed. Moist-labile instability in the atmosphere can be calculated by the amount of latent heat being released during cloud formation, ultimately resulting in increased buoyancy. This process is crucial for cloud formation and precipitation, as the rising air cools and moisture condenses and made it possible to calculate the height up to which cumulus clouds can develop.

Research benefited from the publication of new findings in scientific journals. Most works were translated into other languages and made known to a broad readership.

Hugo Hergesell (1859-1938) was elected as chairman of the Aeronautical Commission founded in 1896 in Paris at the meteorologists' conference of the IMO, and he promoted vertical soundings with kites and pilot balloons in addition to manned balloon ascents (Anonymus, 1909). In 1914 he became head of the Aeronautical Observatory Lindenberg east of Berlin, and from 1922 onwards, nearly daily weather flights from Berlin-Staaken were added. The measurement results were evaluated in great detail.

The above-mentioned cloud type "polar bands" has undergone a change in meaning over the years. A. v. Humboldt assumed that they were divided into parallel strips by electrical processes and would align with the earth's magnetic field towards the magnetic pole (Zöllner, 1871). Around 1870, polar bands received increased attention again. Prestel (1874) described the polar bands as parallel strips of cirrocumulus or cirrostratus, which seemed to converge towards the horizon due to perspective and which often oriented towards the magnetic pole. In 1873, Weber still believed that the earth's magnetic field would influence the direction of the polar bands (Weber, 1873). In Jutland, Sophus Tromholdt (1851-1896) also frequently observed polar bands (Tromholdt, 1873). At the International Meteorologists' Congress in Paris in 1879, it was noted that polar bands were rarely observed in France, but it was confirmed that their observation on the Antilles had repeatedly announced the arrival of

---

[11] Report of the Int. Met. Commission 1907 in Paris, (Berlin, 1908, p. 47).

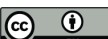



cyclones (hurricanes) (Touchimbert, 1879, p. 505). The convergence point was considered to mark the centre of a storm field. This naturally attracted attention of seamen.

From 1890 onwards, this type of cloud was also referred to as cirrocumuli or altocumuli, even if the strips appeared very long and narrow. Bezold already suggested that waves could form independently of cloud formation, and would then imprint grooves on the stratus cloud. From parallel strips, one can infer the presence of overlying air parcels of different densities that

generate waves when flowing with different velocity. The term thus underwent a change in meaning, as it became clear that the parallel strips were formed by waves. The theoretical explanation of wave formation was developed by Helmholtz (1888). The physicist Adam Paulsen (1833-1907) still held the view in Denmark in 1895 that clouds could be formed by auroras (Paulsen, 1895). He compiled a literature review, suggesting that auroras generate clouds. However, his observations seemed to have been made uncritically.

Later, parallel cirrus bands that appeared to converge at the horizon were then referred to as polar bands (Instructions for Observers at the Weather Observation Stations of the German Reich Weather Service, Reich Weather Service, 1936).

Layer clouds had already been named by Howard. The formation of the cellular structure was theoretically explained by Bénard (1884-1939). He showed that cells (Fig. 14) develop when a critical temperature difference is exceeded between the lower and upper sides of a cloud layer (Bénard, 1900). The upper side cools due to long-wave radiation to space, while the cloud base is

warmed by geoterrestrial radiation. Lord Rayleigh (1842-1919) also developed a formula in 1916 to describe the conditions for these convection patterns (Rayleigh, 1916).

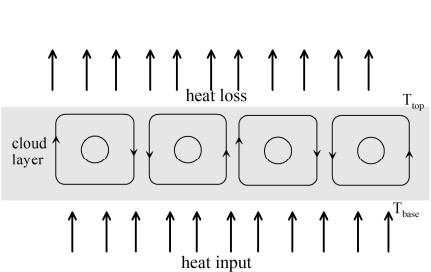
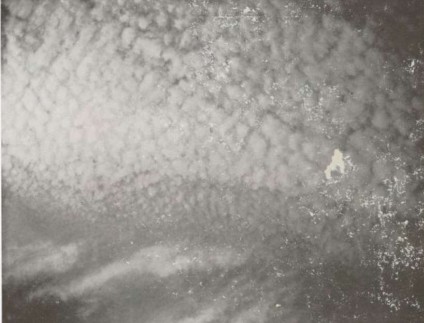

**Fig. 14: Bénard cell formation resulting from exceedance of a critical temperature difference between the lower and upper sides of a layer cloud; right: Altocumulus (Int. Cloud Atlas 1930, pl. 25).**

## 9. Exploration of the third dimension

The exploration of the third dimension was initially carried out using manned balloon flights and improved the understanding of the conditions that are important in forming the cloud shapes. Early flights before 1880, equipped with meteorological instruments, suffered from significant measurement errors. Particularly temperature readings were distorted by radiation biases



(Assmann et al., 1900, p. 15ff.). The barometers were also affected by the acceleration of the balloon. However, flights through
wave clouds allowed for the confirmation that the balloon followed the wave motion, indicating the presence of a wind shear.
A notable experience was gained during a night-time flight by Pierre Testu-Brissy on June 18, 1786, in Paris (Turgan, 1851,
p. 157f.). He encountered a thunderstorm and observed snow crystals and various light phenomena amidst continuous
lightning. At times, he was within a rain cloud, and at other times, within a snow cloud. The light phenomena observed on a
metal tip were tuft-shaped in the rain cloud (St. Elmo's fire) and dot-shaped in the snow clouds. This allowed Testu-Brissy to
conclude on presence of positive or negative electricity (Turgan, 1851, p. 106f.).

The Munich Association for Airship Navigation embarked on its first scientific ascent in 1889, was completing 24 ascents
until 1894 with the meteorologists Finsterwalder, Erk, and Sohnke. The Munich flights were financially supported by the
Bavarian Academy of Sciences through the budget of the Meteorological Central Institute.[12] Isothermal layers and inversions
in the free atmosphere were detected (Fig. 15).

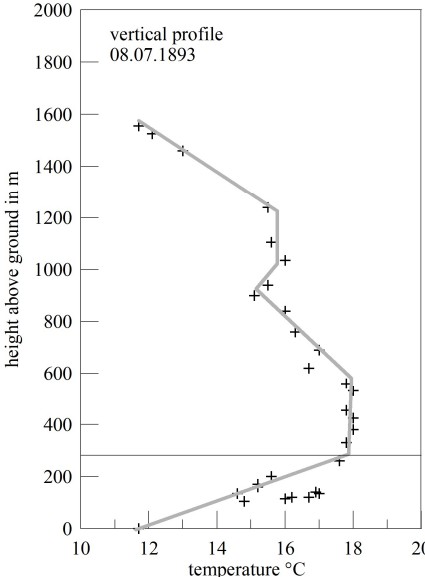


**Fig. 15: Balloon ascent with a vertical profile of the temperature on July 8, 1893 near Munich showing a ground inversion, isothermal layers and an inversion in the free atmosphere. The ascent and descent also provided information of the temporal change in the profile.**

---

[12] Compilation of ascents in: Jahresbericht des Münchener Vereins für Luftschiffahrt 5. Jg, 1894.

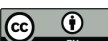

In 1890, another aerological association also began operations in Berlin, conducting 36 ascents between 1890 and 1894.[13]
Assmann and Berson (1900) organized a total of 68 ascents from 1888 to 1896, but altitudes above 6000 m were rarely reached
due to the danger of oxygen deficiency for the pilots. With the establishment of the Aeronautical Observatory Berlin-Tegel in
1900 (Assmann, 1906), later moved to Lindenberg east of Berlin, this branch of research was intensified with 15 balloon
ascents between October 1901 and December 1902. In 1903 and 1904, almost daily ascents with kites or balloons could be

carried out (Dubois, 1993, p. 34). One example of their evaluation of the influence of clouds on the vertical distribution of
moisture is shown in Fig. 16, where cloud type was considered to be the only criterion which would influence the vertical
distribution of moisture. The concept for this evaluation suffered from still missing information of horizontal and vertical
advection of air masses. Horizontal transport may advect more or less moist air, depending on the origin of the airmass
(maritime, continental, subtropical or polar origin), while sinking motion means dry-adiabatic warming of air from aloft in a

high-pressure system or vertical moist-adiabatic lifting in a low combined by the loss of moisture due to precipitation. The
importance of air masses was described first by Jacob Bjerknes in 1919 (see below).

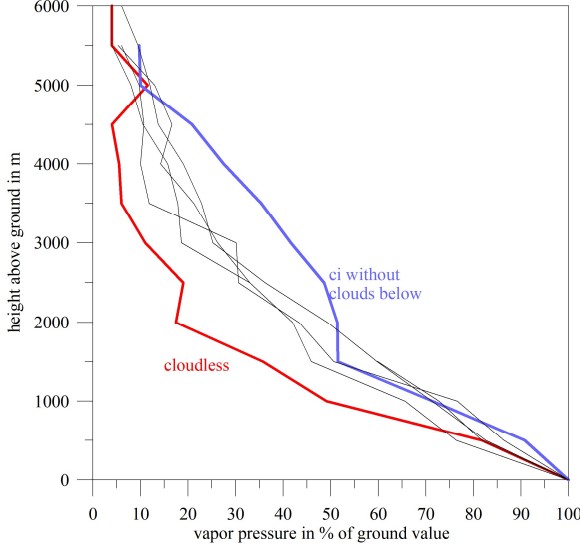

**Fig. 16: Assmann et al. (1990) performed an evaluation of the vertical distribution of moisture. The classification of balloon ascents
was thought to depend on cloud type alone. With now available knowledge, we can interpret the red curve as typical for sinking**

**motion in a high-pressure system, where adiabatically warming enhances the dry situation. The blue curve indicates horizontal
advection of a moist airmass aloft sliding up on cool and dry air near the ground. The three black curves are similar without any
influence of the cloud type being of importance.**

---

[13] Deutscher Verein zur Förderung der Luftschiffahrt, in which military officers and meteorologists cooperated.



In France, the most frequent balloon ascents were arranged, with active centres in Paris, Lyon, and Besancon (a total of 61
ascents in 1904 alone (Aeronautic Club de France, Illustrated Aeronautical Communications, 9, 1905, p. 409-410).

The type of cloud did not play a central role in the first balloon ascents; initially, the focus was on the correct measurement of the vertical distribution of temperature and humidity and comparisons between mountain stations and free atmosphere. There were also experiences with the updraft in cumulus clouds. An early notable observation was that a cloud layer below an inversion is colder and a lot of gas had to be released to bring the balloon to descend through the cloud, otherwise, it would
float on the cloud's upper boundary (Süring, 1895, p. 19). In the next balloon ascents, Süring (1900) found that Clayton's attempt (Clayton, 1896) to treat clouds separately by form, height, and origin was unsatisfactory and differed too much from the internationally agreed classification scheme which should be maintained. Furthermore, Tissandier in France concluded after many balloon ascents that Howard's cloud classification was often unsatisfactory (Tissandier, 1886). The significance of instability and stability of the temperature stratification for the cloud type became apparent only much later, especially with
weather flights performed with planes (Fig. 17).

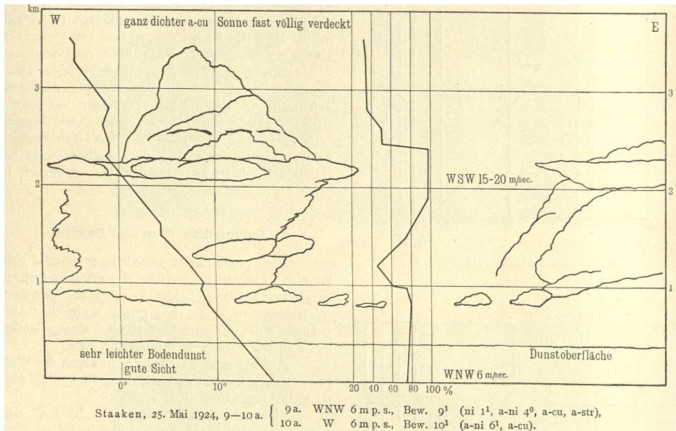

**Fig. 17: Weather flight with vertical profile of temperature and humidity, including indicated cloudiness (Wegener and Schneider, 1926).**

A. Wegener (1911) emphasized in his study on the vertical structure of the whole atmosphere that significant progress had been made through the discovery of inversions in the free atmosphere. While near-surface inversions were already known from the establishment of mountain observatories, such temperature jumps were now detected in the higher atmosphere through balloon flights. Inversions were considered to be boundaries between overlapping air masses with different temperatures, forming a layered structure in the atmosphere (Süring, 1910). Layer clouds spread below an inversion, and Helmholtz waves
or Bénard cells often form at this boundary between air masses of different densities.



"False cirrus clouds" form at the anvil of a thunderstorm, which is the most upper part where the cloud freezes and adopts a streaky, fibrous appearance (E. g. Vincent, 1901). Meanwhile, the term "false cirrus" is out of use, as these clouds are now understood to be real ice clouds.

In 1882, Vettin used a camera obscura to determine cloud movement (Vettin, 1882). In 1888, Mohorovičić also applied this method, complemented by a compass (Mohorovičić, 1888).

In 1890, Hildebrandsson initiated an international program to determine upper-level winds from cloud movements during the International Cloud Year 1896/97. Several devices were developed to accurately measure cloud height (Fig. 18):

- cloud height determination using two-point observations with theodolites;
- use of an aurora borealis theodolite (Föyn, 1900);
- photogrammetry (Koppe, 1896).
-

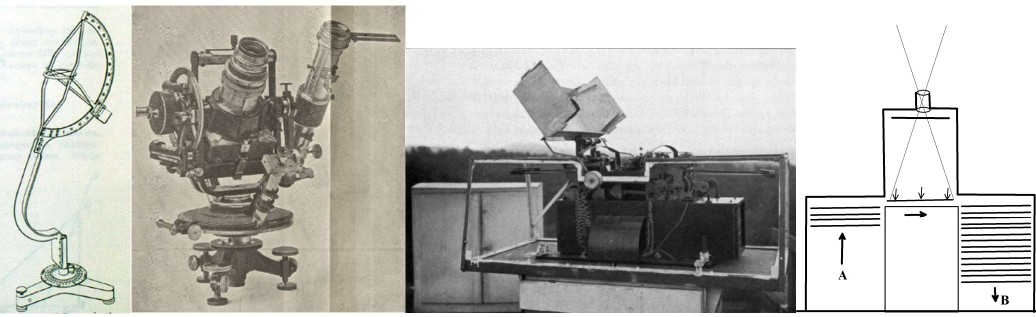

**Fig. 18: Instruments for cloud height and cloud drift measurement include the simple cloud theodolite by Schleich (1932), the photogrammeter by Koppe (1896), and the cloud automat by Sprung (Hellmann, 1912). At the beginning, observations were made at only 13 stations on Earth.**

In Berlin, this imprecise method with two theodolites was further developed into a so-called cloud automat, where two vertically aligned cameras at the ends of a base of about 1500 m could be simultaneously exposed to light by a radio signal, and the photographic plates could be automatically changed. Technically significant was the use of silver-eosin-coated mirror glass plates, which remained usable for weeks (Süring, 1922). Only one person was required for operation of the cloud automat. Consequently, the speed of prominent cloud points could be determined more exactly.

The continuous improvement of these instruments demonstrates the importance of precise observations for understanding the atmosphere motions and weather forecasting.

Initially, observations were conducted at 13 stations on earth using the Koppes photogrammeter. This method was highly labour-intensive, requiring two photographers to coordinate over a distinct cloud point and take simultaneous photographs, with the angles precisely known for trigonometric evaluations. Hildebrandsson (1877, 1892) collected the data and plotted the movement arrows in altitude charts at 4000 m (Fig. 19).

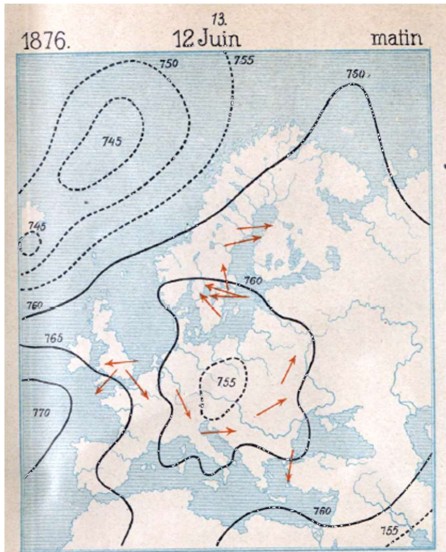

**Fig. 19: Upper-level winds in comparison to isobars at 4000 m altitude compiled by Hildebrandsson (1877).**


Convergences and divergences in the upper atmosphere air flows could be identified by comparisons of upper winds with isobaric maps. It could be concluded that airflow over low-pressure systems is divergent, while it is convergent over high-pressure areas. This helped to infer the vertical movement of air in high and low-pressure systems.

In 1912, a patent was issued for determining the cloud top using a kite (Company Saul, Aachen, report in "Jahrbuch der

Luftfahrt", Vol. 2, 1912, p. 438 f). No systematic observations on results have been found.

Abercromby published a proposal for cloud classification in 1887 titled "Suggestions for an international nomenclature of clouds," which he had developed together with Hildebrandsson (1838-1925), one of the leading cloud researchers. [14] Abercromby travelled around the world twice to verify that the same cloud types appeared everywhere (Abercromby, 1887b).

Besson described in 1902 an erroneous determination of the horizontal cloud notion with a nephoscope in cases when an

updraft or downdraft in a cloud was established (Besson, 1902).

Wehrlé and Schereschewsky (1923) described an idealised cloud scheme in a typical atmospheric low pressure system in 1923 (as depicted in Fig. 11).

The theoretical conditions for the formation of cloud rolls under specific wind shear conditions was published by Kuettner (1971), but they were not given a special name.

---

[14] He was the first to describe the ENSO phenomenon (El Nino – Southern Oscillation).



In 1901, Süring pointed out the existence of moist layers at preferential altitudes in the atmosphere over Berlin, identified from averaging 68 balloon ascents (Süring, 1901). It was concluded that cloud formation preferentially occurs in these layers (Fig. 20). Whether this result could be generalized, was never checked.

In 1918, Köppen noted in a brief communication that one of the most urgent tasks in meteorology was to link cloud forms with the results from aerological ascents. Obviously, there were still deficits in an unambiguous determination of cloud forms.

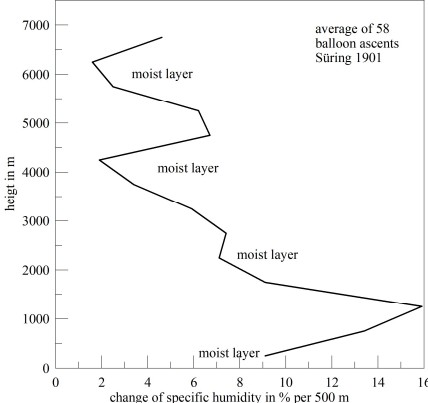


**Fig. 20: Mean profiles of the vertical temperature gradients over Berlin, from which moisture layers became visible in which clouds preferentially develop.**

## 10. Cloud Atlases

Bezold (1894) stated in 1894 that he could not explain the large variety of forms of cirrus clouds (wind trees) and why only
occasionally the influence of waves was clearly visible. Even until 1922, the diversity of cirrus cloud forms had not been standardized. A revised International Cloud Atlas and a unified international nomenclature for naming clouds were the result of many meetings and internationally coordinated cloud research. The final nomenclature was established at the 6th Directors' Conference in Copenhagen in 1929.

The inadequacy of photography in representative cloud photos was a serious problem for a long time. Initially, photography
was completely insignificant because clouds could hardly be distinguished from the blue sky on photographic plates. A first improvement was achieved using yellow filters (Hildebrandsson, 1889). Riggenbach (1889) in Switzerland recognized the possibility of using mirrors in cloud photography. Since the blue sky emits strongly polarized light and clouds emit only weakly polarized light, the reflected image on the photographic plate becomes high-contrasting (Wolf, 1993, p. 14f). Thus, the compilation of a cloud atlas using photographs was difficult.

Weilbach in Denmark created an atlas with 16 of his personally painted colour illustrations (Weilbach, 1881).

In 1890, a first international Atlas des Nuages appeared with painted cloud images in French, English, and German language (Hildebrandsson et al., 1890).



In Munich, a first classification of cloud types was agreed upon in 1891, and an Atlas Committee was formed (members were: v. Bezold (Prussia), Bilwiller (Switzerland), Davis (Argentina), Hann (Austria), Hepites (Romania), Hildebrandsson (Sweden), Mascart (France), Mohn (Norway), Paulsen (Denmark), Scott (England), Snellen (Netherlands), and Tacchini (Italy)). In 1892, a call was published in the Meteorologische Zeitschrift (Vol. 27, p. 80) and other journals, asking artists and amateurs to send in cloud images to Committee members so that an appropriate selection could be made based on scientific criteria. In the May issue of the same journal, it was reported that usable cloud photographs had been received. A Munich art institution had produced 12 trial plates using the "light printing technique", published in an atlas by Singer (1892), with new designations deviating from those of Howard.

In 1894 Bezold addressed the inadequacy of photography for another time, despite progress had been made using yellow filters. In the same year 1894, the "Atlas Committee" met in Uppsala (Rotch, Blue Hill, USA), Teisserenc de Bort (Trappes, France), Weilbach (Denmark), Broounof (Russia/Ukraine), Fineman (Sweden), Hagström (Sweden), Riggenbach (Switzerland), and Sprung (Berlin)). They selected suitable images for the international atlas from an exhibition of 300 photographs taken in different countries. In 1896 the second International Cloud Atlas, edited by Riggenbach, Hildebrandsson, and Teisserenc de Bort, was published, along with observer instructions (Hildebrandsson, 1896). The images had been heavily edited, as cloud images were combined with different foregrounds to achieve a realistic impression, and retouching was done (Wolf, 1993, p. 40). Nevertheless, the light printing technique still proved insufficiently.

For the meteorological service in USA, Clayton published in 1897 an atlas "Illustrative Cloud Forms" with 16 colour plates (Clayton, 1897).

In 1899 Polis published a cloud atlas in Aachen with 16 plates (Polis, 1899).

In 1902, Captain Wilson-Barker in London suggested a complete collection of cloud types, similar to taxonomies in botany, geology, and zoology, during the reorganization of the Royal Meteorological Society (Wilson-Barker, 1902). He evidently expected a greater diversity than the types agreed upon from a scientific standpoint so far.

In 1903, Vincent in Belgium published an atlas: Etudes sur les nuages with 6 images (Vincent, 1903).

In 1910, the second edition of the International Cloud Atlas was published with 14 colour plates. This atlas was still deemed insufficient by the German Antarctic Expedition of 1911/12. The photos were of poor quality, requiring extensive personal experience for using it successfully (Barkow, 1924, p. 56).

This compilation, while not exhaustive, demonstrates the intensive international efforts to optimize depictions of various cloud types. Pharmacist Ernst Mylius (1846-1929) in Berlin, an amateur sailor at the Baltic Sea, had asked numerous shipman and sea pilots about their understanding of weather signs from cloud formations but received unsatisfactory answers. Consequently, he started painting watercolour pictures of clouds in relation to weather development, limiting himself to the clouds observed in his vicinity (Mylius, 1906). He considered photography unsuitable, as camera opening angles did not match the human viewing field, and he deemed colour perception essential. His realistic cloud paintings were exhibited at the 1906 International World's Exhibition in Milan (Mitteil. Dt. Seefischerei, 1909, p. 353).

As the light printing technique used in the 1896 cloud atlas was generally deemed inadequate, Köppen and Alfred Wegener contacted Mylius in 1917, requesting watercolours for a new cloud atlas. Mylius provided a collection of 400 watercolour paintings.[15] However, due to World War I and the poor economic situation in Germany, the project did not come to fruition.

In 1944, naval meteorologist Hans Frank (Mylius, 1944) once again brought attention to Mylius' paintings and compiled a cloud atlas with 34 plates from the images provided by Mylius' son. It was accepted that no images of foehn clouds or mountain clouds were available. Ernst Mylius had an excellent perception of colour and he observed how the colour of clouds depended on vegetation and its stages of development throughout the year. Allegedly, the atlas was intended to assist German small combat units in World War II in recognizing weather conditions during military missions. This view must be questioned by the fact that the images were accompanied by only brief meteorological explanations (Fig. 21).

---

[15] Correspondence in: Archiv Deutsches Museum, München, Nachlass A. Wegener.


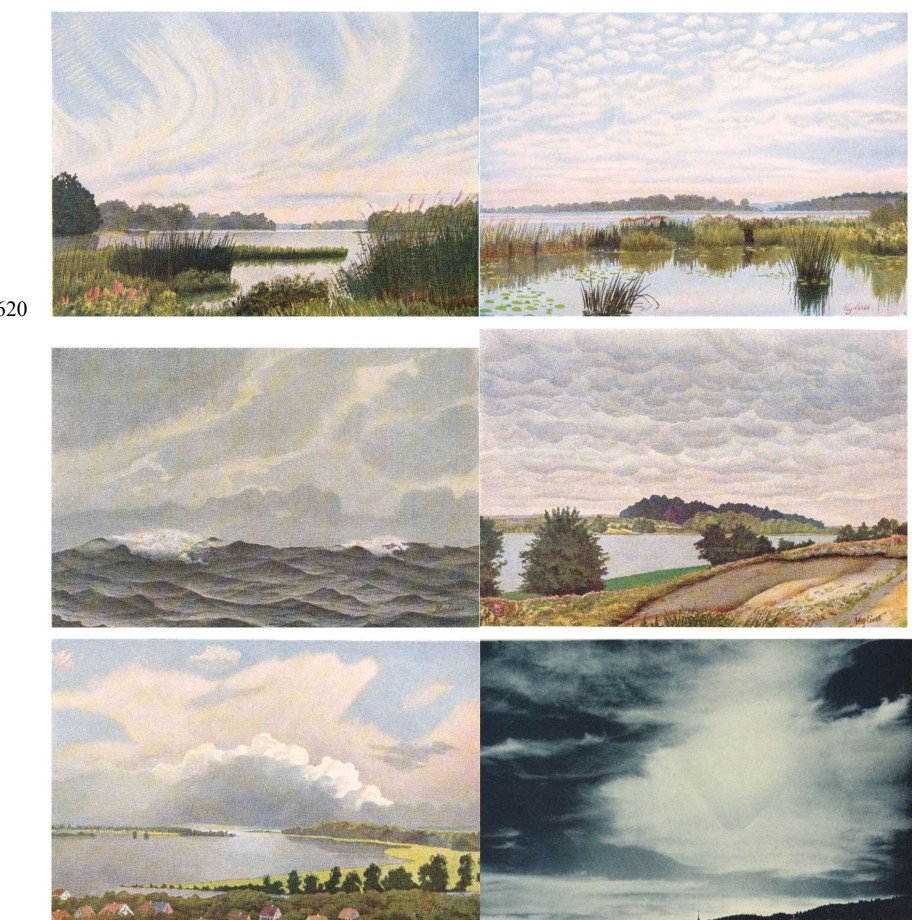

**Fig. 21: Selection of watercolour clouds from the Mylius-atlas (1944). For comparison a black and white cloud photo by Quervain (1908) is added, demonstrating the still poor quality of cloud photographs.**


In 1920, a cloud atlas by Clarke with 40 plates was published in London (Clarke, 1920) and in 1921, the Meteorological Office in London published another atlas entitled "Cloud Forms."

In 1924, a cloud atlas of the USA was published containing 32 illustrations. Accompanying this was a graphic depicting the altitude ranges of various cloud types (Fig. 22).



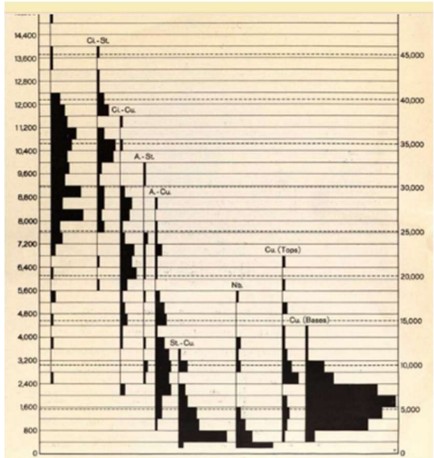


**Fig. 22: Height ranges of various clout types (USA Cloud Atlas, 1924).**

In 1925, Fontseré published a cloud atlas for Catalonia, which contained 32 plates (Fontseré, 1925).

In 1926, Durr and Wehrlé published a cloud atlas in Paris with 59 plates. The photographs were still not particularly brilliant

and for this reason they were supplemented with drawings and explanations (Fig. 23).



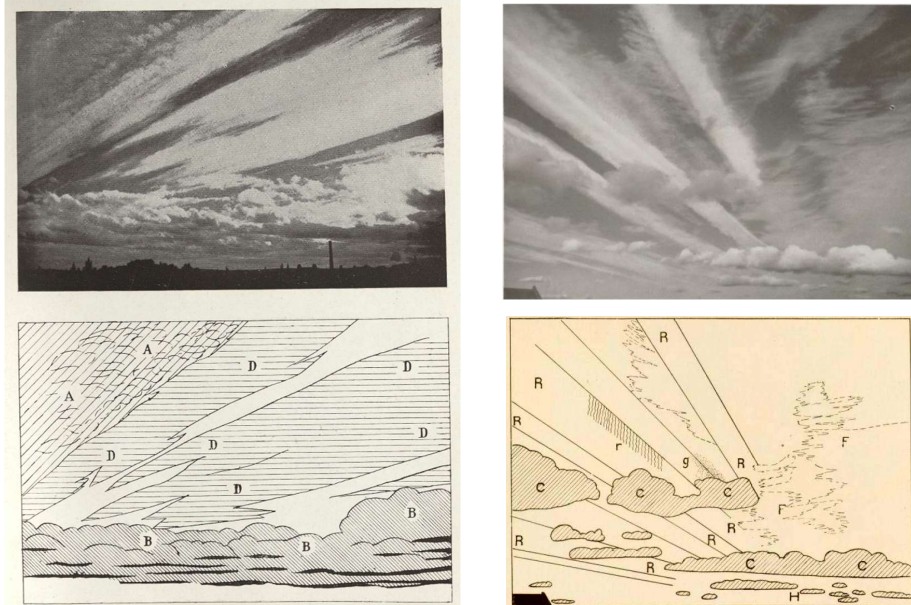

**Fig. 23: Left top and bottom: Example of cirrus with cumulus and explanation from the French Atlas (Durr and Wehrlé, 1926), pl. XLVII. Right: Example of cirrus radiatus ("R" polar bands) with cumulus ("C") from the International Atlas of Clouds (1939, Pl. 11).**

In other national meteorological services, parallel developments occurred where local cloud types were specially depicted. However, it was necessary to learn to look beyond local conditions and recognize the essential characteristics. By today's standards, the cloud photographs of that time were generally of poor quality.

A new International Commission for the Study of Clouds was set up in 1921 and produced the International Atlas of Clouds and of Types of Skies in 1932, proceeded by an abridged edition in 1930 for use by observers to meet the requirements of coding changes. New editions of both publications followed in 1939 (International Commission for the Study of Clouds, 1932, 1939).

A new International Cloud Atlas was published in two volumes in 1956 after the foundation of the World Meteorological Organization (WMO) in 1951 (WMO, 1956).

4000


**11. Present day's valid types and characteristics of clouds**

The International Cloud Atlas of 1956 distinguishes 10 cloud genera and 15 species (Table 2). The genera are differentiated by altitude and by stratiform or convective types. In each altitude range, the types develop specific appearances, characterized primarily by convection within a limited altitude range. Stable or unstable air stratification is crucial for shaping the clouds.

655 During overcast conditions, stable stratification prevails, but an overlaying slow uplift generates structureless forms. Radiative influences in stratiform clouds induce instability and create Bénard cells. If the atmosphere is unstable throughout the altitude range, thunderstorms (cumulonimbus clouds) will form.

**Table 2 Cloud nomination**

| | 10 Genera | 15 Species | 9 varieties | supplementary features |
|---|---|---|---|---|
| 660 | Cirrus | fibrates | intortus | incus |
| | Cirrocumulus | uncinus | vertebratus | mamma |
| | Cirrostratus | spissatus | undulatus | virga |
| | Altocumulus | castellanus | radiatus | cavum |
| 665 | Altostratus | floccus | lacunosus | fluctus |
| | Nimbostratus | stratiformis | duplicatus | asperitas |
| | Stratocumulus | nebulosus | translucidus | praecipitans |
| | Stratus | lenticularis | perlucidus | arcus |
| | Cumulus | volutus | opacus | murus |
| 670 | Cumulonimbus | fractus | | tuba |
| | | humilis | | cauda |
| | | mediocris | | |
| | | congestus | | |
| | | calvus | | |
| 675 | | capillatus | | |

It should be noted that the mentioned species and varieties do not occur in combination with all genera. For example, cumulus capillatus forms when a rapidly growing cumulus triggers condensation in an overlaying moist layer upon rapid rising. They are rare and difficult to photograph. Quervain (1908) therefore created a drawing (Fig. 24).



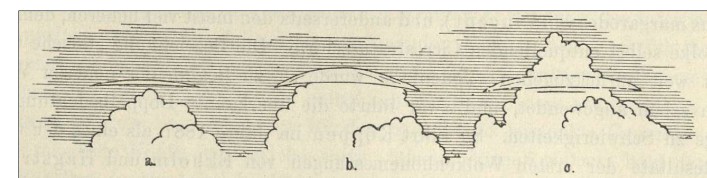


**Fig. 24: Drawing of a cumulus capillatus by Quervain (1908), showing the development with time.**

## 12. Stratospheric and mesospheric clouds

While most clouds form in the troposphere, clouds can occur under special conditions in the stratosphere or mesosphere. These are not necessarily pure water clouds. Not all observers initially determined their height correctly. Stratospheric clouds can

form over mountains at altitudes between 20 and 29 km (Fig. 25) and occur mainly in winter when the polar vortex is pronounced. When air flows over mountains, waves are created, and clouds form at the wave crests. These clouds contribute to ozone depletion, meaning they have a strong atmospheric chemical effect. Mohn referred to them as Mother-of-pearl clouds (Norwegian: Perlemorskyer; Mohn, 1893). In recent times, they were renamed as Polar Stratospheric Clouds (Fig. 26a). Mohn did not distinguish between polar stratospheric clouds and noctilucent clouds and calculated their height much too high to be

over 100 km. Hildebrandsson (1895) criticized the term "Mother-of-pearl clouds" as unclear since iridescent clouds occur at many altitudes, especially in the troposphere, but he confirmed their great height. Störmer correctly determined their height to be 21-27 km (Fig. 25) (Störmer, 1932, 1940).

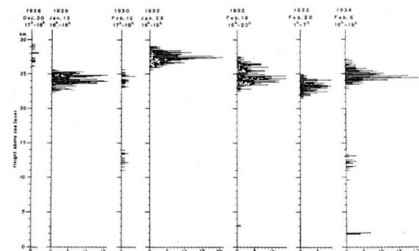

**Fig. 25: Altitude determinations of Mother-of-pearl clouds in Norway.**


Mesospheric clouds form only in summer at relatively high latitudes at altitudes between 80 to 89 km, as the lowest temperatures in the atmosphere occur at this time at the mesopause. They are termed as noctilucent clouds (Fig. 26b) and were first described in detail by Jesse in 1887, who later determined their height very accurately (Jesse, 1885, 1886, 1887, 1896). Further sightings were also reported in 1887 (Grützbach, 1887). They are visible only in the evening or at night when

illuminated by the sun standing below the horizon. Mohn (1893) calculated their height to be 140 km; Jesse disagreed in 1893 but was unaware of the existence of polar stratospheric clouds in the lower range near 20 km (Jesse, 1893). After Jesse's first

report, the open question was discussed in the Berlin Academy of Sciences whether it could be volcanic ash from the 1883 eruption of Krakatau. According to later estimates, the volcanic ash from this eruption rose to maximally 38 km (Assmann, 1887, p. 275).

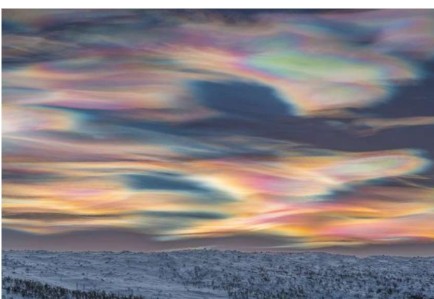
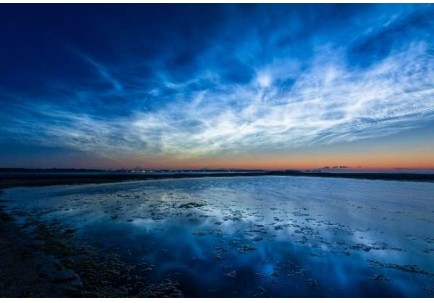


**Fig. 26a: Polar stratospheric cloud (Finland, credit: Thomas Kast; downloaded from), 26b: Noctilucent cloud (Laboe, 2019, June 21<sup>th</sup> credit: Matthias Suessen).**

### 13. Anthropogenic activities and cloud formation

Contrails have been often termed as cumulus homogenitus that could only form once aircraft were capable of flying at high
altitudes. The first reports date back to 1919: Ettenreich published an observation from 1915 in Tyrol, Austria, which had been withheld due to wartime censorship regulations (Ettenreich 1919). On June 21, 1919, during a record high-altitude flight exceeding 8000 m, a contrail was detected in Munich (Weickmann, 1919), who published his observation (Fig. 27). In the USA, Wells (1919) described the contrail of a German fighter aircraft in World War I in 1918, which became famous as the "Argonne battle cloud." Alfred Wegener (1920) described the formation process in his essay "Cirrus and Ice Supersaturation"
in which he emphasized the different vapour pressure over water and ice, which leads to the evaporation and disappearance of supercooled droplets and the transformation into a pure ice cloud.

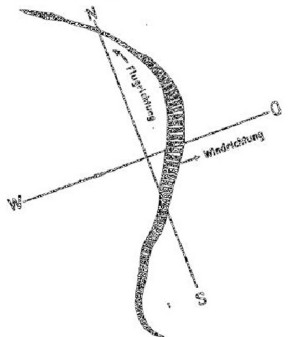

**Fig. 27: First drawing of a contrail observed in Munich in 1919 (Weickmann, 1919) with indication of direction of flight and wind.**

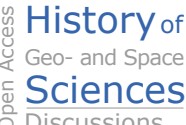

During World War II, contrails revealed the locations of aircraft (Baucom 2007). As such, it was crucial to identify the meteorological conditions that are required for their formation in order to avoid areas prone to contrail development. Contrails form when the water vapour emitted by engines leads to supersaturation. Ongoing studies on contrails have presented notable work, such as the internationally recognized research by Schmidt (1940). He investigated technical methods to reduce the likelihood of contrail formation, including the use of hydrogen-poor fuels and the mixing of cooler and dryer air into the

exhaust.

Intense fires can trigger cumulus cloud formation under unstable atmospheric conditions, known as cumulus pyrogenitus or cumulus flammagenitus. Eaton (1893) described cloud formation over a reed-burning site in winter in California and suggested a closer investigation of the conditions under which large fires could trigger cumulus formation at an inversion. In his situation moist-unstable air from the sea flowed inland above the inversion. When the convection induced by the fire was strong enough

to break through the inversion a cumulus could form. Similarly, Ward (1897) described cumulus formation originating from the smoke plume of a fire near Boston, USA (Fig. 28), under conditions of moist-unstable maritime air flowing inland. At that time, the conditions for moist instability were not yet well understood. Ward referred to Espy's work, who had described the enhancement of instability through the release of condensation heat. Another report on cumulus formation during intense fires near Milton, Massachusetts, was published by Ferguson and Brooks (1899).


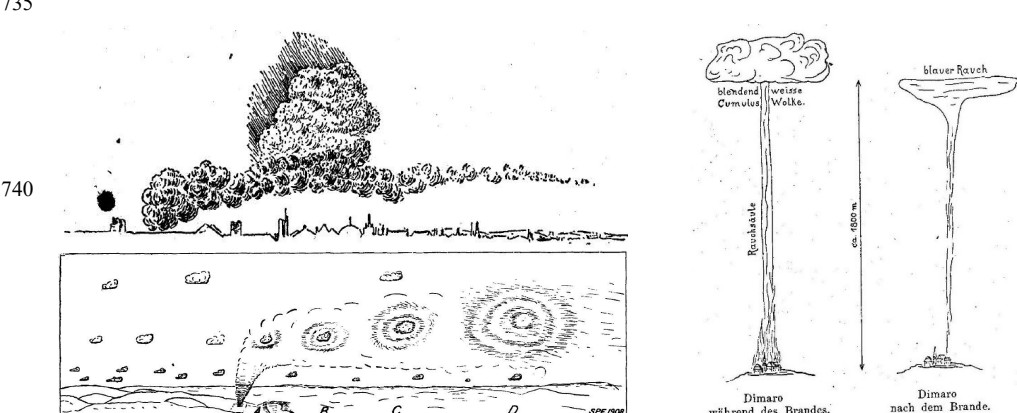


**Fig. 28, a) Drawings of cumulus clouds, that formed over intensive fires: above after Ward (1897); b) after Ferguson (1899); and c) after Ettenreich (1919).**


Another anthropogenic cloud type is termed fumulus. Fumulus clouds can form over intense heat and water vapour sources, such as power plants with cooling towers.

An astonishing anthropogenic modification effect on a cloud was observed during World War II: in this case a halo in a cirrus cloud was seen in England in 1944, where sound waves from war activities (cannon fire) created rapidly moving bright and
dark streaks in the associated ice clouds forming the halo (Archenhold, 1944).

## 14. Use of Satellites

Cloud images from weather satellites have provided new insights into larger-scale cloud structures that were previously unobservable from a ground-based single location. For example, Fig. 29 shows the formation of Karman vortices behind a Cape Verde Island in the Atlantic Ocean. Additionally, the spiral frontal pattern around the centre of a low-pressure system
near Iceland shows details of the mixing process of warm and cold airmasses which could not be recognized in this form from ground-based cloud observations. It was recognized that the Bergen cyclone model was only a coarse approach depicting the real situation incompletely.

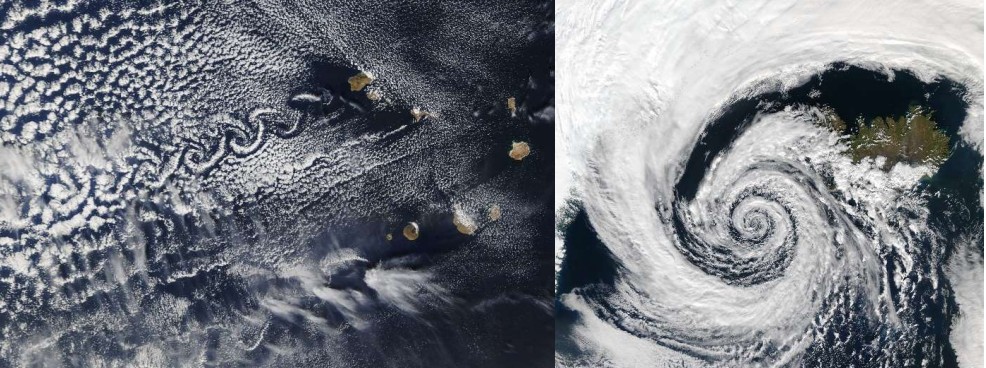

**Fig. 29: Karman vortex street in the lee of the Canary Islands and low-pressure vortex near Iceland, questioning the completeness**
**of the Concept of the Bergen cyclone model (credit: EumetSat).[16]**

## 15. Conclusion

Recent studies have corroborated that clouds significantly contribute to global heat transport through the release of latent heat. Helmholtz had already stated this in 1888, arguing that heat conduction, in comparison to radiation and convective transport, could only account for negligible heat fluxes and was significant only at interfaces such as the earth's surface and at internal
boundaries of different air masses (Helmholtz, 1888, p. 331). Further confirmation came from Stüve (1888-1935), who observed that the potential temperature of air masses at high altitudes remains relatively stable over weeks, whereas it varies significantly in low air masses also due to cloud formation (Stüve, 1922). Cumulus clouds of varying thicknesses release latent

---

[16] Downloaded 2025, February 25.



heat at the corresponding altitudes, but individual clouds have a short lifespan. Moreover, the ascent processes in extratropical cyclones (warm conveyor belts) contribute most significantly to global energy transport toward the poles through cloud

formation (Binder and Madonna, 2020). Stratus clouds, with their extensive coverage, are long-lived and inhibit the earth's longwave outgoing radiation. Reaching these insights required an extensive study of cloud types, with research becoming more intensive and internationally coordinated from around 1880 onwards. It was found that typical cloud types occur globally when the formation conditions are met. Understanding cloud forms was only fully achieved through studying the three-dimensional structure of the atmosphere, including the stability and instability of temperature stratification and the vertical distribution of

water vapour. Observing cloud movement provided insights into large-scale convergences and divergences and the resulting ascent and subsidence processes in atmospheric pressure systems. The depiction of cloud types in atlases significantly contributed to the standardization of cloud nomenclature, although technical capabilities were inadequate for long time. At first, paintings were preferred because they better corresponded to human perception and were preferred due to the early limitations of photography.

The scientific investigation of cloud formation conditions has led to the refinement of initially proposed cloud classifications, which emphasized some form too much and thereby generated an excessive number of cloud types. Theoretical studies have also contributed to understanding certain cloud structures. Knowledge of specific cloud formations allows for insights into atmospheric conditions and, to a limited extent, predictions of weather development. In many countries, meteorological observations have been automated meanwhile, meaning that visual observation of cloud types and their changes is no longer

performed.

Intense discussions in scientific journals have been invaluable in achieving an international agreement on cloud names. Without this exchange, progress in understanding the atmospheric physical conditions would have been much slower, and misinterpretations would have persisted. The understanding of cloud formation processes is still ongoing. Recent studies (Bluestein 2024) indicate that while the basic cloud forms are understood, modern measurement techniques such as radar and

lidar continue to provide further insights into cloud formation. Understanding the atmosphere as a medium with chaotic behaviour has initiated first investigations on a fractal behaviour of clouds (Christensen et al., 2021). Tornadoes, which are not a cloud form, can arise from mesoscale, highly unstable cloud systems that exhibit rotation. Many cloud formations result from specific local conditions (e.g., "Moazagotl cloud" in the Riesengebirge, Germany), "cloth of Table Mountain", "morning glory" (Australia) which is a sort of bore cloud formed at the collision point of sea breeze with stable inland air (Christie,

1992). These forms were not included in the international nomenclature and cloud atlases, and thus such local trivial names persist. Howard's first attempt to introduce a Latin nomenclature has achieved complete acceptance on the way to the international nomenclature.




**16 Code availability**

No codes have been used.

**17 Data availability**

Only data available in the literature have been used. These have been cited in the references.

**18 Author contribution**

There is only myself as sole author.


**19 Competing interests**

The author declares that he has no conflict of interest.

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
