# Peer review of "History of Research on Cloud Types and Naming of Clouds"

_History of Geo- and Space Sciences, 2025_

## Referee Comment (RC1)

**Paper review for HGSS – August 2025**

***History of Research on Cloud Types and Naming of Clouds***

By Peter Winkler

*MS ref. hgss-2025-3*

**GENERAL COMMENTS**

A useful review of the historical development in the taxonomy of cloud types, but it is much too long (47 pp typescript, around 12 500 words excl references). It would benefit enormously from restructuring (suggestions below) and thorough editing to reduce the MS length to max 8000 words before references (which should also be trimmed only to those relevant and strictly necessary).

In this reviewer's opinion there is also – not surprisingly, given the author's nationality – a greater emphasis on Germanic works and authors. The balance could be helped by reading and considering inclusion some of the English-language works suggested in my review below. However, any suggestions for inclusion must not be allowed simply to increase the length of the piece; careful considerable reconsideration of which topics to include and which others to omit is essential.

My recommendation would be 'Major revision needed'. I would be happy to consider reviewing a revised paper.

**SPECIFIC COMMENTS**

The author appears to have attempted both a loosely chronological timeline together with subject headings, but the range of topics makes this approach extremely cumbersome and very difficult to follow in places. My suggestion would be to consider editing and restructuring the work by collecting related studies under appropriate headings: for example, these could be (in no particular order) –

- Cloud droplet properties (visual and dynamical)
- Development and understanding of cloud forms
- Assessing then measuring cloud altitudes
- Orography
- Electricity and cloud formation/development
- Clouds as forecasting tools
- Vertical distribution of temperature and humidity
- Satellite era

- Development of cloud atlases
- Etc

The point is made several times that early instruments, photography etc were insufficient for accurate work. Of course; but the point is over-laboured in my view. We cannot expect Luke Howard writing in 1802 to have detailed knowledge of cloud dynamics, or access to high-resolution photography, but clearly he and others laid the foundations for today's understanding even without such tools and techniques. Thus line 40-41 is clearly at question in this respect: "Extensive atmospheric studies using instruments were necessary to achieve a uniform nomenclature of cloud shapes" when Howard pointed out the main shapes (heap = convective, layer = dynamical uplift and ice clouds) and began to build a workable taxonomy around them over 200 years ago – work since has built on Howard's foundations, rather than replaced them.

Equally, there are numerous instances in the paper referring to work that was later proven to be incorrect, or work published that simply had little impact, then or later. Most if not all of these could be omitted to reduce length. Examples: Line 210-215 and Fig 6, lines 437-439, Lines 514-515, and others.

**TECHNICAL CORRECTIONS etc**

I have not attempted to list every sticking point – my review would be 5000 words long if I did so. Instead, I list below some typical inaccuracies/inconsistencies/duplicated matter that should be checked more carefully, and/or edited down or omitted altogether.

Abstract refers (line 24) to 'the modern International Cloud Atlas, published in 1956'. It would surely be preferable to refer to the latest WMO Cloud Atlas published online in 2018.

There are some minor points of English that should be looked at. Line 28 'upgliding' (dynamical ascent?); line 164 'cloud drops' (water vapour?), line 301 'unstable air stratification' (instability), line 356 "The evolution of cloud observation for weather forecasting was fascinating forever " (??), line 605 'shipman' (mariners?), 'Anonymus' in the references ('anonymous'), etc.

Superscripted footnotes make the text more difficult to read, and in places are confused with cited references. Review/edit into citations with notes where appropriate, then retain only where absolutely necessary. Some entries include birth and death dates and these is some confusion between these and multiple reference citations in the same section of text.

Line 50: "artistic representations do not contribute to scientific research". I would most certainly dispute this, starting with Howard's beautiful watercolours c 1805, and John Constable in the 1820s. A recent book by Edward Graham, *Clouds: How to identify*

*nature's most fleeting forms*.  Princeton University Press, 2025, shows cloud types as painted by artists, mostly C19 or C20, and is worth reading with this sentence in mind.

Line 65: IPCC, IAMAS – define acronyms. IPCC in 1955?

Section 2: 'Fantasy on clouds' … 'comparable to childhood'. This section could be shortened to a paragraph. Similarly, section 3 could be edited down (I also find it odd that the author states Line 99 that 'the scientific community [the word had not been invented by then, of course] did not pay attention to his [Robert Hooke's] suggestions', when Hooke's suggested methods for taking and tabulating observations were widely adopted, well into C21.

Line 110-120 contradiction: Halley was unaware of Guericke's experiments – despite a review appearing in *Phil Trans*, at a time when Halley was an active member? Unlikely!

Line 37: Kratzenstein 1743-44 and air molecules – the concept of molecules was yet many decades ahead, while the composition of 'air' was not understood for decades (Priestley's discovery of oxygen, 1774 and subsequently: Argon was not discovered until 1894, the third-largest constituent of atmospheric air). Check and reword. Also, subsequently Line 146: there is hardly 'almost 200 years' between Kratzenstein conclusions in 1740s and their being questioned (1805, according to Line 151).

Line 139: Diffraction was first written about by Grimaldi in 1660, it was not 'discovered' by Fraunhofer in 1825.

Line 169: "Clouds are constantly changing their forms that make air movements visible". I think most would say this is the other way around.

Line 179: "The Societas Meteorologica Palatina focused only on observing cloud coverage, density, and colour". No – they published many years of instrumental records. They can hardly be criticised for a lack of cloud type observations when no agreed taxonomy yet existed.

Line 190: the abrupt statements that "Howard's significant achievement was the introduction of Latin terms …" and (line 199) "Howard did not yet understand the stable and unstable states of stratification" are a huge disservice to Luke Howard. His main achievement was recognising some commonality in cloud forms, and positing their development from one type to another (such as small cumulus into Cb over time). His or any other proposed Latin taxonomy would have been of little benefit without this major insight. Suggested reading: Hamblyn, R., 2001: *The Invention of Clouds: How an amateur meteorologist forged the language of the skies*.  New York: Farrar, Straus and Giroux.

Line 260-266: Ley's book was published posthumously in 189**4** (line 266) yet line 260 gives his lifetime as 1840-189**6**? Which is correct? Line 480-485 and Fig 16: Assman ref to 1990, should be 1890? Also, Fig 16 shows Ci at 2000 m? Fig 19 dated 1877 yet text suggests

1892 citation? Line 551 refers to Fig 11 'in 1923', when Fig 11 is dated 1939. There are other mistakes of this type in the MS: edit and check carefully.

Line 290ff: there were many manned and instrumented balloon flights in England from the 1860s, many by James Glaisher and co-workers: there are many accounts in the meteorological literature  for example, Marriott, W., 1903: James Glaisher, F.R.S. 1809-1903. Quart. J. Royal Meteorol. Soc., 29, 115-121. Hunt, J. L., 1978: James Glaisher, FRS (1809-1903). Weather, 33, 242-251. Hunt, J. L., 1996: James Glaisher FRS (1809-1903) Astronomer, Meteorologist and Pioneer of Weather Forecasting: 'A Venturesome Victorian'. Quart. J. Royal Astronomical Society, 37, 315-347. Includes 131 references

Line 312: the word 'pilots' is used – does the author mean this in the sense of balloonists or aircraft pilots, with the different timelines this implies?

Line 336: "At the meteorological congress in Munich in 1891, the first cloud classification was agreed upon based on the suggestions of Abercromby and Hildebrandsson (Abercromby, 1887a)." This both duplicates text, and is out of place – I suggest it would fit better in a section covering the development of cloud atlases.

Section 5, Prognostic use: Consider also the very first chapter of Napier Shaw's *The drama of weather* (Cambridge University Press, first edition 1933) started with using clouds as forecasting tools. Such approaches were very important for forecasting, at least at local/regional levels, alongside the development of conceptual models such as Bjerknes and the Norwegian School, until the adoption and increasing sophistication of computer models. This section could usefully bring together and edit down related and duplicated topics elsewhere in the MS.

There is also no mention of cloud physics textbooks, such as those by Mason and Ludlam, and particularly the development of the tephigram (or skewT diagram) by Napier Shaw from the 1920s.The tephigram is relevant also around line 725, as the printed British Met Office version of the tephigram include a dashed line labelled 'MINTRA', which is the threshold T/Dew point/Pressure for contrail formation. Tephigrams also very easy to explain and demonstrate stability, instability, potential temperature and many other inferences essential to understanding cloud dynamics.

Fig 13: I am unable to determine the point of this image. Many others could be chosen – for example, the pioneering work in early aircraft by British forecaster CKM Douglas from 1917 (see Sutcliffe, R. C., 1982: Obituary - C K M Douglas 1893-1982. Quart. J. Royal Meteorol. Soc., 108, 996-997. doi 10.1002/qj.49710845825.) or the many examples, photographs and beautifully printed (in black and white) in CJP Cave's *Clouds and weather phenomena* (Cambridge University Press, 1926). Line 569 states "The inadequacy of photography in representative cloud photos was a serious problem for a long time" but even a casual examination of Cave's book shows that this was not true at least by 1926. The description and examples from the 'Mylius atlas' of 1944 confuse this point rather than clarify.

Section 10 Cloud atlases: Might make an interesting paper by itself. But there is too much content here – reduce and simplify to the more significant publications, listing only the most relevant others in a short table rather than in text (perhaps list country, language, year, author/publisher, page count, photographs/watercolours, black/white or colour plates, etc).

It seems odd not to include in this section more recent publications – for example WMO Cloud Atlas (online) in 2018, UK Met Office *Cloud types for observers* (1982), Gavin Pretor-Pinney's *The Cloudspotter's Guide* (Hodder & Stoughton / Sceptre, London, 2006), *The Cloud Collector's Handbook* (Hodder & Stoughton, 2009), and *A cloud a day* (Batsford Press London, 2019), in addition to the many (often beautifully illustrated) non-specialist books on cloud identification.

Fig 22: I am unable to determine the point of this image, suggest omission.

Section 12: Stratospheric and mesospheric clouds. While certainly of interest, I wonder whether this topic might be more usefully covered as a 'Part 2' paper, shortening Part 1 and allowing it to focus on tropospheric clouds?

Section 14, satellites: mention might be made here of Harry Wexler's foresight that satellite imagery would be of crucial importance – see for instance his chapter in Fleming, J. R., 2016: Inventing atmospheric science: Bjerknes, Rossby, Wexler and the foundations of modern meteorology. MIT Press Cambridge, Mass., 2016 with a very early suggestion (c1955 I think) of what a 'satellite image' might look like.

*Stephen Burt*

*Department of Meteorology, University of Reading, UK*

*8 August 2025*